# The Role of Intraperitoneal Intraoperative Chemotherapy with Paclitaxel in the Surgical Treatment of Peritoneal Carcinomatosis from Ovarian Cancer—Hyperthermia versus Normothermia: A Randomized Controlled Trial

**DOI:** 10.3390/jcm11195785

**Published:** 2022-09-29

**Authors:** Angela Casado-Adam, Lidia Rodriguez-Ortiz, Sebastian Rufian-Peña, Cristobal Muñoz-Casares, Teresa Caro-Cuenca, Rosa Ortega-Salas, Maria Auxiliadora Fernandez-Peralbo, Maria Dolores Luque-de-Castro, Juan M. Sanchez-Hidalgo, Cesar Hervas-Martinez, Antonio Romero-Ruiz, Javier Briceño, Álvaro Arjona-Sánchez

**Affiliations:** 1Oncologic and Pancreatic Surgery Unit, University Hospital Reina Sofia, Avda Menéndez Pidal s/n, 14004 Córdoba, Spain; 2CIBERehd, Maimonides Biomedical Research Institute of Cordoba (IMIBIC), University Hospital Reina Sofia, Avda Menéndez Pidal s/n, 14004 Córdoba, Spain; 3Department of Pathology, Reina Sofía University Hospital, 14004 Córdoba, Spain; 4Maimonides Biomedical Research Institute of Córdoba (IMIBIC), 14004 Córdoba, Spain; 5Department of Analytical Chemistry, Campus of Rabanales, University of Córdoba, Annex Marie Curie Building, 14071 Córdoba, Spain; 6Department of Computer Science and Numerical Analysis, University of Córdoba, 14071 Córdoba, Spain; 7Department of Biochemistry and Molecular Biology, University of Córdoba, 14004 Córdoba, Spain

**Keywords:** ovarian cancer, peritoneal carcinomatosis, intraperitoneal chemotherapy

## Abstract

Background: The treatment of ovarian carcinomatosis with cytoreductive surgery and HIPEC is still controversial. The effect and pharmacokinetics of the chemotherapeutics used (especially taxanes) are currently under consideration. Methods: A phase II, simple blind and randomized controlled trial (NTC02739698) was performed. The trial included 32 patients with primary or recurrent ovarian carcinomatosis undergoing cytoreductive surgery (CRS) and intraoperative intraperitoneal chemotherapy with paclitaxel (PTX): 16 in hyperthermic (42–43 °C) and 16 in normothermic (37 °C) conditions. Tissue, serum and plasma samples were taken in every patient before and after intraperitoneal chemotherapy to measure the concentration of PTX. To analyze the immunohistochemical profile of p53, p27, p21, ki67, PCNA and caspase-3 and the pathological response, a scale of intensity and percentage of expression and a grouped Miller and Payne system were used, respectively. Perioperative characteristics and morbi-mortality were also analyzed. Results: The main characteristics of patients, surgical morbidity, hemotoxicity and nephrotoxicity were similar in both groups. The concentration of paclitaxel in the tissue was higher than that observed in plasma and serum, although no statistically significant differences were found between the two groups. No statistically significant association regarding pathological response and apoptosis (caspase-3) between both groups was proved. There were no statistically significant differences between the normothermic and the hyperthermic group for pathological response and apoptosis. Conclusions: The use of intraperitoneal PTX has proven adequate pharmacokinetics with reduction of cell cycle and proliferation markers globally without finding statistically significant differences between its administration under hyperthermia versus normothermia conditions.

## 1. Introduction

The standard treatment of primary advanced epithelial ovarian cancer (EOC) is complete cytoreductive surgery (CRS) with no residual tumor, followed by adjuvant chemotherapy based on taxanes and platinum compounds [1,2].

Due to its natural history, EOC remains localized in the abdominal cavity. Intraperitoneal (IP) chemotherapy allows higher concentrations of chemotherapeutics in the peritoneal cavity than systemic chemotherapy because of its slow absorption by the peritoneum and the first-pass effect by the liver. This allows higher IP doses and potentially increased efficacy with concurrently low systemic toxicity [3,4] and an increased possibility of survival. Nevertheless, this type of chemotherapy has not been fully accepted for different reasons: pharmacokinetic problems (the penetration depth of IP delivered drugs into the tumor nodules is limited [5], so optimal CRS will be required before IP chemotherapy), problems with the technique since it is not free from peritoneal access device complications with high toxicity [6] and the complex and demanding logistical management of patients. Although IP chemotherapy with taxanes has been demonstrated to be effective in advanced EOC [6], in the last GOG-252 trial no progression-free survival (PFS) improvement with IP chemotherapy was seen [7].

To overcome the inconveniences of IP chemotherapy, intraoperative administration under hyperthermia conditions (HIPEC) arose. Its main objective was to treat residual microscopic disease after CRS before the formation of adhesions, through physical (heat) and chemical (chemotherapeutic) methods [8]. The mechanisms by which HIPEC results in an increased tumor response to cytostatics, besides the direct effect of heat [9] per se, are multiple [10]. HIPEC potentiates the cytotoxic effect of some chemotherapeutic agents [11,12,13,14,15] and increases their tissue penetration [16,17]. There is significant heterogeneity in fundamental aspects of HIPEC administration, such as the clinical setting in which it is indicated, the definition of optimal surgery (CC1 vs. R1), the cytostatic used and its dose, the temperature (37–46 °C) or the perfusion time [18].

Some authors [19,20,21,22,23,24,25,26,27] (our group among them) use taxanes for HIPEC in the treatment of ovarian carcinomatosis because of its high efficacy observed in the systemic treatment of EOC and its favorable pharmacokinetics after IP administration due to its high molecular weight [28] and hepatic metabolism. The theory about an increase in the efficacy of intraperitoneal taxanes administration is supported by different clinical [29,30,31,32] and experimental studies [33,34]. In an update published by Sugarbaker [35], it was observed that hyperthermia increases the cytotoxic activity of most cytostatics. However, this synergy is not clear in taxanes. Contradictory results have been obtained concerning the interaction of heat with taxanes [36], even though they are heat stable, and hyperthermia seems to increase the intracellular accumulation of these cytostatics.

For all the above, this study aims to analyze the effect of intraoperative IP administration of paclitaxel (PTX) under hyperthermia vs. normothermia conditions on antitumor activity, proliferation and cell cycle markers and its pharmacokinetics.

## 2. Patients and Methods

A phase II, simple blind, randomized controlled trial (RCT), NTC02739698, was performed. All steps, including a selection of patients, sampling and storage, were developed according to the guidelines dictated by the World Medical Association Declaration of Helsinki in 2004. The ethical review board of Reina Sofía Hospital (Córdoba, Spain) approved and supervised the clinical study.

## 3. Inclusion Criteria

Age ranging between 18–75 years; histopathologic confirmation of peritoneal carcinomatosis from primary or recurrent EOC (stage IIIb-IIIc FIGO); Karnosfsky index > 70 or Gynecologic Oncology Group performance status ≤ 2; optimal or complete CRS (no residual tumor greater than 2.5 mm) and an informed consent form filled by all patients.

## 4. Exclusion Criteria

Unfulfillment of inclusion criteria; extra-abdominal metastasis or stage IV FIGO; concomitance of other malignant neoplasm; renal, hepatic or cardiovascular dysfunction; intolerance during treatment or refusal to participate.

## 5. Sample Size Calculation

Based on an expected 40% of G3 tumor-regression in the experimental arm vs. a 1% of G3 tumor-regression in the control arm with an α error of 0.05 and β error of 0.20, the sample size was 32 (16 patients per group). It has been calculated according to the available funding from the public health grant. 

## 6. Treatment

All patients (except one) received a neoadjuvant chemotherapy regime consisting of four to six cycles of carbo-taxol. After confirming the stabilization or regression of the disease, the patients underwent optimal CRS followed by intraoperative IP chemotherapy with 60 mg/m^2^ PTX per 2 l of 1.5% dextrose at continuous perfusion for 60 min. They were randomized 1:1 after the completeness of cytoreduction into two groups: the experimental arm (H-group) in which the IP chemotherapy was administered in hyperthermia conditions (41–42 °C) and the control arm (N-group) where this IP chemotherapy was administered in normothermia conditions (37 °C). After surgery, most patients received adjuvant carbo-taxol chemotherapy to complete the eight cycles.

## 7. Variables

Main characteristics of patients, the ovarian cancer stage, previous surgical score (PSS) [37], response to neoadjuvant chemotherapy (complete response: normalization of Ca 125 and disappearance of signs of disease in radiology tests; partial response: decrease of the value of Ca 125 and decrease of signs of disease in radiology tests according to RECIST criteria [38]) and data of surgery such as peritoneal carcinomatosis index (PCI) [39] or Completeness of Cytoreduction (CC) Score [40] were collected.

Dindo–Clavien scale [41] and CTCAE [42] v 4.0 were used to describe surgical morbidity and hematological and renal toxicities, respectively. Major morbidity was considered as ≥grade 3 after 30 days from the surgical treatment.

## 8. Sampling and Storage

Two types of peritoneal biopsies (with and without tumor) were taken before and after IP chemotherapy (PRE-chemo and POST-chemo, respectively). Those with a small and well-perfused area of infiltrated peritoneum were sent to the Pathology Department in fresh. The other tumor-free peritoneal biopsies and blood samples (taken before, immediately after and 1 h after IP chemotherapy-PRE-chemo, POST-chemo and 1 h POST-chemo, respectively), the latter after centrifugation-were introduced in Eppendorf tubes and preserved at −80 °C to analyze the concentration of PTX in peritoneal tissue by liquid chromatography-tandem mass spectrometry (LC–MS/MS), as reported before [43].

## 9. Anatomopathological Study

The specimens were fixed in 10% neutral formalin, routinely processed and embedded in paraffin blocks, from which 3 micrometer (μm) thick serial sections were cut and stained with HE, PAS and Masson’s trichome.

The immunohistochemical study was performed using the prediluted antibodies: p53 (clone DO.7. Dako Corporation Santa Clara, CA United States), p27^kip1^ (clone DSC-72. Genova Scientific SL), p21^waf1^ (clone polyclonal. Genova Scientific SL, Seville, Spain), ki67 (clone MIB.1. Dako Corporation), PCNA (clone PC10. Genova Scientific SL, Seville, Spain) and caspase-3 (clone polyclonal. Master Diagnostica SL). For the immunostaining, the Dako EnVision Flex Plus visualization system was used. The sections were examined by two blinded expert pathologists and evaluated by the grouped Miller and Payne (MP) system [44] for pathological response: G1 (minimal changes that include MP G1-G2), G3 (microscopic foci that include MP G3-G4) and G5 (no residual tumor), according to the percentage of total area involved in the biopsy specimen. Immunohistochemical expression was assessed by the percentage of nuclei-stained tumor cells.

## 10. Statistical Analysis

Continuous variables were expressed as means and standard deviation (SD), and categorical variables as frequency and percentages. Association between categorical variables was tested using Pearson’s chi-squared test (χ^2^). The difference between means of continuous variables was tested by an independent *t*-test. The second set of dependent *t*-tests (paired samples *t*-test) was carried out to compare the means of two related groups to determine if a statistically significant difference exists between these means of anatomopathological results. *p* values ≤ 0.05 were defined as statistically significant.

## 11. Results

### 11.1. Patients’ Main Characteristics

The main patient characteristics are presented in Table 1. No statistically significant differences in preoperative variables were found between the two groups, except BMI (24.68 ± 2.55 H-group vs. 29.00 ± 4.84 N-group, *p* < 0.004).

### 11.2. Treatment and Morbidity

The mean PCI was similar in both groups, and although microscopically complete cytoreduction (CC0) was achieved in 81.2% in H-group and 62.5% in N-group, no significant differences were found (Table 2).

There was no treatment-related death. However, all patients in this study had at least one grade 2 surgical complication since they all received total parenteral nutrition (TPN), and most patients required a blood transfusion.

Major surgical morbidity (≥ IIIa) was 25% in H-group and 12.5% in N-group. In the H-group, two patients had a wound infection that needed surgical debridement; one had low-grade colorectal fistula treated with conservative treatment and percutaneous drainage, and the other required reintervention due to hemoperitoneum. In N-group, one patient required reoperation due to hemoperitoneum, and another had septic shock with reintervention with renal and global respiratory failure, resulting in death.

Grade 3–4 hemotoxicity was seen in 9.4% (12.5% H-group vs. 6.3% N-group). Grade 3–4 nephrotoxicity was seen in 12.5% of the H-group and in 25% of the N-group (Table 2).

## 12. Pharmacokinetics

PRE-chemo serum, plasma and tissue samples had PTX values below the detection limit. In the H-group the mean PTX concentration in serum post-chemo and 1 h post-chemo was 16.61 ± 5.34 ng/mL and 12.18 ± 5.52 ng/mL, respectively; in plasma post-chemo and 1 h post-chemo, it was 17.24 ± 6.14 ng/mL and 11.23 ± 5.15 ng/mL, respectively; and in tissue, post-chemo, it was 1382 ± 1407.18 ng/mL. In the N-group, the mean PTX concentration in serum post-chemo and 1 h post-chemo was 14.98 ± 4.79 ng/mL and 13.37 ± 4.87 ng/mL, respectively; in plasma post-chemo and 1 h post-chemo, it was 15.14 ± 6.18 ng/mL and 12.36 ± 4.87 ng/mL, respectively; and in tissue post-chemo, it was 2093.19 ± 1777.92 ng/mL. No significant differences were found between the two groups in any measurement. However, it was observed that the concentration of PTX obtained at the local level (tissue) was much longer than the systemic (plasma and serum) in both groups (Figure 1).

## 13. Anatomopathology

Regarding the pathological response according to MP grouped system, no significant differences were observed in either group. It was observed that IP chemotherapy produced a marked reduction of tumor cellularity in 87.5% of the H-group and 81.3% of the N-group. The pathological reductions GR1, GR2 and GR3 were observed in 12.5%,62.5% and 25% for HIPEC group and 18.8%, 62.5% and 18.8% for the normothermic group (*p* = n.s) (Table 3). No significant differences concerning apoptosis (caspase-3) were found either.

The analysis of the results of the cell cycle markers (p53, p27 and p21) showed that there was a significant reduction in the expression ofW the three markers after IP chemotherapy in the 32 patients (*p* = 0.021, *p* = 0.000 and *p* = 0.000, respectively) (Figure 2), but when both groups were compared, this reduction was not statistically significant. Something similar occurred with cell proliferation markers (ki67 and PCNA). After comparing pre- and post-IP-chemo samples globally, the differences were statistically significant (*p* = 0.012 and *p* = 0.000), but not when pre- and post-chemo samples from both groups were compared (Table 3). No statistically significant differences were observed between the two groups for the cell cycle and proliferation markers.

## 14. Discussion

The present study has not shown statistically significant differences in regression grade, pharmacokinetic or molecular markers when the PTX was administered intraperitoneally in normothermia vs. hypethermia conditions. However, our study found that PTX is an excellent drug to be used intraperitoneally independent of hyperthermia conditions.

Numerous worldwide medical centers have incorporated CRS with peritonectomy procedures associated to HIPEC to treat peritoneal carcinomatosis, making this technique controversial when the carcinomatosis originates from the colon [45,46]; however, nowadays such treatment is the standard care in pseudomyxoma peritonei [47,48] and mesothelioma [49]. Although the standard treatment of ovarian carcinomatosis is not CRS-HIPEC [1,2], evidence of its use is growing after recent publications of RCTs [50,51,52,53] and meta-analyses [54,55].

Spiliotis et al. [50] reported an improvement in survival in the treatment of recurrent EOC with CRS-HIPEC vs. CRS alone, where the mean overall survival (OS) was 26.7 vs. 13.4 months respectively. However, this study has limitations considering the randomization process and the definition of the end points, which affect the interpretation of the results [56]. Moreover, others [57,58] have raised the concern that the statistical analysis performed in the study was not clearly described and inappropriately applied and their recalculation of statistics demonstrated no statistically significant differences between the two groups.

For primary EOC, better disease-free survival (DFS) rates and OS were observed in patients treated with neoadjuvant chemotherapy (NAC) followed by interval CRS-HIPEC, compared to those treated with NAC followed by interval CRS alone [51,52,53].

The first meta-analysis [54] of CRS-HIPEC in EOC concludes that this combination improves OS rates for both primary and recurrent EOC vs. isolated CRS. However, this meta-analysis did not provide the exact pooled hazard ratios associated with HIPEC in the clinical setting. A later meta-analysis [55] concluded, for primary EOC, that CRS-HIPEC improved both DFS and OS (in patients with a residual tumor ≤ 1 cm, while not visible tumors improved DFS but not OS). In case of recurrent EOC, CRS-HIPEC improved OS but not DFS (in patients with residual tumor ≤ 1 cm or not visible tumor improved DFS, while in patients with residual tumor ≤ 1 cm only improved OS).

Our group has carried out the CRS-HIPEC with PTX in the treatment of ovarian carcinomatosis since 1997 [59]. However, in the beginning, it was not always possible to use the perfusion machine that allowed reaching hyperthermia, and IP chemotherapy was administered in normothermia conditions observing how these patients reached similar conditions of survival. This behavior, added to the contradictory results obtained concerning the interaction of heat with taxanes [35], led us to decide to perform the present study.

The analysis of the results showed two homogeneous groups according to pre- and peri-operatives variables, except BMI, which was significantly higher in the N-group. However, this difference did not affect either group’s ability to achieve optimal cytoreduction, which is consistent with the literature as well [60]. The morbidity and mortality outcomes of CRS-HIPEC were similar to the literature [61,62], with the total significant morbidity of 25% in the H-group and 12.5% in the N-group. Although twice as much, no significant differences were found between the two groups. Major morbidity related to intraperitoneal PTX, such as hematological and renal toxicity, ranged from 10.5% to 84.2% and 0% to 7%, respectively [6,63,64]. For HIPEC PTX administration, the major hematological toxicity is reported from 0% to 13% [22,65] and renal toxicity above 11,6% [66]. In our study, we observed significant hematological complications in 12.5% of the H-group and 6.3% of the N-group and major renal toxicity in 12.5% of the H-group and 25% of the N-group, which were not statistically different.

In our study, the maximal tissue concentrations were average, 84.54 and 178.01 times longer than the maximal plasma concentration (H and N-group, respectively). This fact supports the idea that PTX could be suitable for IP administration, according to previous reports of intraperitoneal use of PTX [22,67]. Although PTX concentration in the N-group was almost twice that of H-group, this fact might be related to the contradictory results of the effect of hyperthermia on the pharmacokinetics of taxanes [36], and no significant differences were observed in both groups.

To assess the effect of IP administration of PTX on the pathological response (reduction in tumor cellularity) we used the grouped Miller and Payne system, widely studied in the effect of neoadjuvant on locally advanced breast cancer [68] but not used previously in the treatment of ovarian cancer with HIPEC. Although we did not find significant differences in our study in the two groups, it was observed that IP chemotherapy produced a marked reduction of tumor cellularity in 87.5% of the H-group and 81.3% of the N-group.

Proliferation and cell cycle control are central processes in the biology of cancer [69], and our study showed for the first time the effect of HIPEC on these biomarkers. As p53 mutation seems to be related to the development of chemoresistance and recurrence [70,71], the relevance of the expression p21 and p27 as prognostic survival factors in EOC are inconsistent [72,73,74,75]. Our findings showed a reduction in the expression of the three markers after IP chemotherapy in both groups, but no statistically significant differences were found among them. That shows the possible influence of IP chemotherapy in the molecular field of EOC. More studies are needed to assess these findings. In an experimental setting, the hyperthermia with IP PTX used in mice with EOC enhanced the antitumor effects through immune-mediated cancer stem cell targeting [76]. In an experimental study from De Bree et al. [77], the absence of thermal enhancement (the normothermic), as our study showed, may be as effective as hyperthermic intraoperative intraperitoneal chemotherapy with taxanes. Potential oncological and treatment-related adverse effects of concurrent hyperthermia, such as thermal injury to organs and other tissues [78], immunosuppressive effects [79] and the enhanced systemic release of heat-shock proteins [80,81,82], could be avoided.

The main limitation of this study is the small sample size, even though our hospital is a reference for the treatment of ovarian carcinomatosis in our country, so it will be necessary to conduct additional studies with a larger sample size to validate the impact of the temperature in IP administration of PTX on pharmacokinetics, pathological response and cell cycle markers. Nevertheless, to our knowledge, this is the first study in ovarian carcinomatosis where IP chemotherapy is compared with taxanes in hyperthermia vs. normothermia conditions.

In conclusion, in this clinical trial, PTX has proven to have adequate pharmacokinetics to treat ovarian carcinomatosis, reaching optimal concentrations in tissue and minimal in serum and plasma, as well as a reduction in cell cycle and proliferation markers globally when administered in the peritoneal cavity during CRS. Nevertheless, no significant differences in pharmacokinetics and cytotoxicity could be demonstrated between normothermic and hyperthermic intraoperative intraperitoneal chemotherapy in patients with primary or recurrent ovarian cancer.

## Figures and Tables

**Figure 1 jcm-11-05785-f001:**
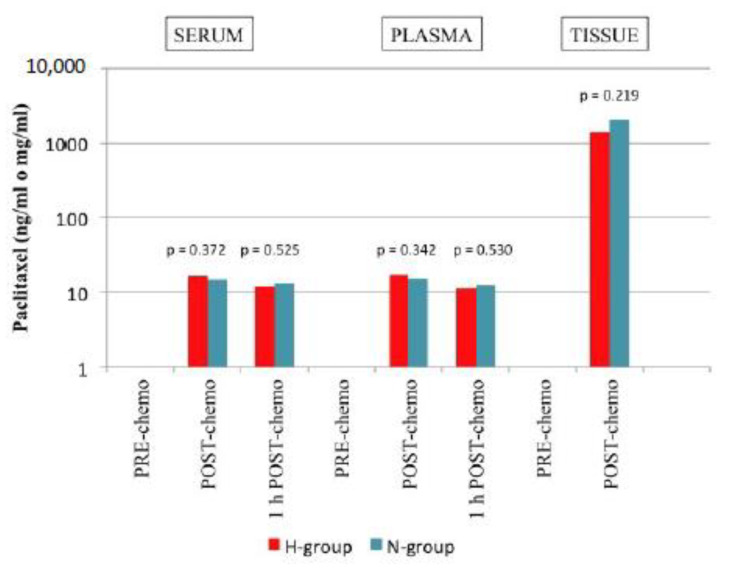
Pharmacokinetics of intraperitoneal PTX administration in our study. The concentration of PTX in the tissue (local level) was higher than that observed in plasma and serum (systemic level), although no statistically significant differences were found between the two groups.

**Figure 2 jcm-11-05785-f002:**
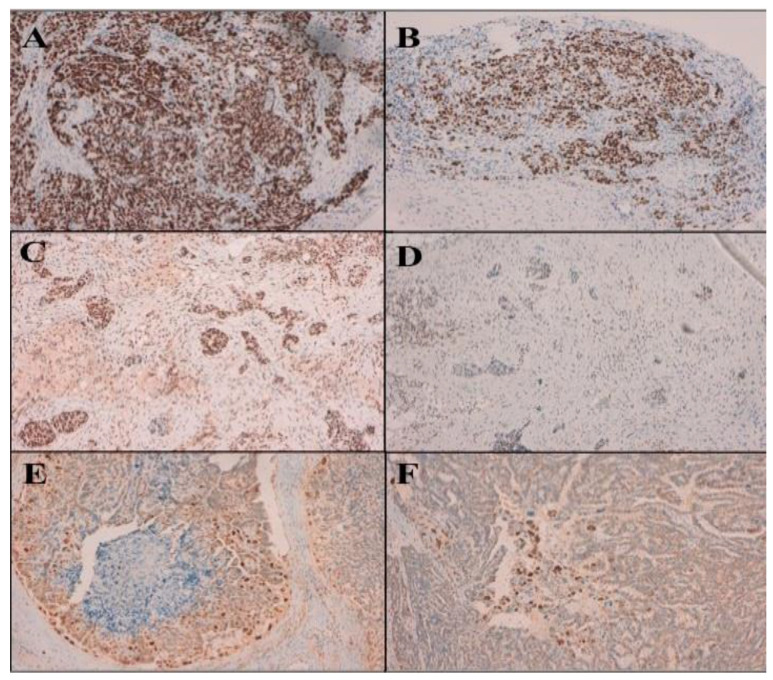
Immunohistochemical nuclear expression of cell cycle regulatory proteins in one H-group patient. In the analysis of the expression of the cell cycle regulatory proteins (p53, p21 and p27) in this patient, it is observed that in the pre-chemo samples the positive nuclear labeling (brown staining) is much more abundant than in the samples post-chemo; that is, there is a significant reduction in its expression after chemotherapy. (**A**): p53 pre-chemo, (**B**): p53 post-chemo, (**C**): p27 pre-chemo, (**D**): p27 post-chemo, (**E**): p21 pre-chemo, (**F**): p21 post-chemo.

**Table 1 jcm-11-05785-t001:** Patients’ main characteristics.

	H-Group	N-Group	*p*
	(*n* = 16)	(*n* = 16)	
Age (years)	57.06 ± 12.60	58.13 ± 9.38	0.789
BMI (kg/m^2^)	24.68 ± 2.55	29.00 ± 4.84	0.004
Prior abdominal surgery			0.253
No	13 (81.2%)	9 (56.2%)	
Yes	3 (18.8%)	7 (43.8%)	
Prior comorbidity			1
No	8 (50.0%)	9 (56.3%)	
Yes	8 (50.0%)	7 (43.8%)	
Ovarian cancer situation			0.33
Primary	15 (93.8%)	12 (75.0%)	
Recurrent	1 (6.3%)	4 (25.0%)	
Prior cancer surgery			0.075
No	10 (62.5%)	4 (25.0%)	
Yes	6 (37.5%)	12 (75.0%)	
Prior Surgical Score (PSS)			0.164
0	10 (62.5%)	4 (25.0%)	
1	4 (25.0%)	8 (50.0%)	
2	2 (12.5%)	3 (28.8%)	
3	0 (0.0%)	1 (6.3%)	
Neoadjuvant chemotherapy			0.31
No	1 (6.3%)	0 (0.0%)	
Yes	15 (93.8%)	16 (100%)	
Response to neoadjuvant chemotherapy			
Complete			0.714
Partial	2 (12.5%)	4 (25.0%)	
	13 (81.2%)	12 (75.0%)	

H-group: Hyperthermia group, N-group: Normothermia group, BMI: Body Mass Index.

**Table 2 jcm-11-05785-t002:** Treatment and morbidity.

	H-Group	N-Group	*p*
	(*n* = 16)	(*n* = 16)	
Ureteral catheterization			0.544
No	2 (12.5%)	1 (6.3%)	
Yes	14 (87.5%)	15 (93.8%)	
PCI	19.25 ± 6.78	21.50 ± 7.81	0.391
Peritonectomy procedure			0.462
Total	11 (68.8%)	12 (75.0%)	
Extensive	5 (31.3%)	3 (18.8%)	
Pelvic	0 (0.0%)	1 (6.3%)	
CC Score			0.432
CC0	13 (81.2%)	10 (62.5%)	
CC1	3 (18.8%)	6 (37.5%)	
Splenectomy			0.703
No	12 (75.0%)	10 (62.5%)	
Yes	4 (25.0%)	6 (37.5%)	
Number of anastomosis			0.363
0	5 (31.3%)	8 (50.0%)	
1	10 (62.5%)	6 (37.5%)	
2	1 (6.3%)	2 (12.5%)	
Number of procedures			0.665
2–1	0 (0.0%)	1 (6.3%)	
4–3	11 (68.8%)	10 (62.5%)	
5	5 (31.3%)	5 (31.3%)	
Intraoperative blood transfusion			0.154
No	5 (31.3%)	9 (56.3%)	
Yes	11 (68.8%)	7 (43.8%)	
Duration of surgery (minutes)	492.53 ± 95.81	538.06 ± 112.91	0.237
Postoperative blood transfusion (units)			0.688
0			
1–2	4 (25.0%)	1 (6.3%)	
3–4	7 (43.8%)	7 (43.8%)	
>4	3 (18.8%)	5 (31.3%)	
	2 (12.5%)	2 (12.5%)	
Postoperative stay (days)	12.38 ± 6.63	13.33 ± 6.65	0.691
Surgical Morbidity (Clavien)			0.245
I-II	12 (75.0%)	14 (87.5%)	
IIIa	3 (18.8%)	0 (0.0%)	
IIIb	1 (6.3%)	1 (6.3%)	
IVa	0 (0.0%)	1 (6.3%)	
IVb	0 (0.0%)	0 (0.0%)	
V	0 (0.0%)	0 (0.0%)	
Leukopenia (CTCAE 4.0)			1
No	14 (87.5%)	15 (93.8%)	
1	0 (0.0%)	0 (0%)	
2	0 (0.0%)	0 (0%)	
3	2 (12.5%)	1 (6.3%)	
4	0 (0.0%)	0 (0.0%)	
5	0 (0.0%)	0 (0.0%)	
Neutropenia (CTCAE 4.0)			0.491
No	14 (87.5%)	15 (93.8%)	
1	0 (0.0%)	0 (0.0%)	
2	0 (0.0%)	0 (0.0%)	
3	1 (6.3%)	1 (6.3%)	
4	1 (6.3%)	0 (0.0%)	
5	0 (0.0%)	0 (0.0%)	
Thrombocytopenia (CTCAE 4.0)			0.478
No	15 (93.8%)	13 (81.2%)	
1	0 (0.0%)	1 (6.3%)	
2	1 (6.3%)	2 (12.5%)	
3	0 (0.0%)	0 (0.0%)	
4	0 (0.0%)	0 (0.0%)	
5	0 (0.0%)	0 (0.0%)	
Acute kidney injury			0.346
(CTCAE 4.0)			
No	11 (68.8%)	6 (37.5%)	
1	2 (12.5%)	2 (12.5%)	
2	1 (6.3%)	4 (25.0%)	
3	2 (12.5%)	3 (18.8%)	
4	0 (0.0%)	1 (6.3%)	
5	0 (0.0%)	0 (0.0%)	
Hematuria (CTCAE 4.0)			0.931
No	8 (50.0%)	7 (43.5%)	
1	2 (12.5%)	2 (12.5%)	
2	6 (37.5%)	7 (43.8%)	
3	0 (0.0%)	0 (0.0%)	
4	0 (0.0%)	0 (0.0%)	
5	0 (0.0%)	0 (0.0%)	

H-group: Hyperthermia group, N-group: Normothermia group, PCI: Peritoneal Carcinomatosis Index, CC score: Completeness of Cytoreduction Score, CTCAE: Common Terminology Criteria for Adverse.

**Table 3 jcm-11-05785-t003:** Anatomopathological results.

	H	N	H + N	*p*
	(*n* = 16)	(*n* = 16)	(*n* = 32)	
Pathological response				0.842
GR1 (minimal changes)	2 (12.5%)	3 (18.8%)		
GR3 (microscopic foci)	10 (62.5%)	10 (62.5%)		
GR5 (no residual tumor)	4 (25%)	3 (18.8%)		
p53				
ΔPre-Post H + N				0.021
ΔPre-Post H vs. Pre-Post N	21.56 ± 39.99	11.75 ± 37.89	16.66 ± 38.64	0.482
p27				
ΔPre-Post H + N				0
ΔPre-Post H vs. Pre-Post N	59.38 ± 38.55	44.38 ± 29.15	51.88 ± 34.47	0.224
p21				
ΔPre-Post H + N				0
ΔPre-Post H vs. Pre-Post N	36.25 ± 38.28	23.31 ± 34.02	29.78 ± 6.22	0.32
Ki67				
ΔPre-Post H + N				0.012
ΔPre-Post H vs. Pre-Post N	11.19 ± 24.32	8.88 ± 18.68	10.03 ± 21.36	0.765
PCNA				
ΔPre-Post H + N				0
ΔPre-Post H vs. Pre-Post N	47.38 ± 44.18	23.25 ± 37.49	35.31 ± 42.13	0.106
Caspasa-3				
ΔPre-Post H + N				0.188
ΔPre-Post H vs. Pre-Post N	23.44 ± 49.35	−2.81 ± 32.96	10.31 ± 43.38	0.089

H: hyperthermia, N: normothermia.

## Data Availability

Data available contacting with first or corresponding authors previously.

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
