# Peer review of "The Role of Intraperitoneal Intraoperative Chemotherapy with Paclitaxel in the Surgical Treatment of Peritoneal Carcinomatosis from Ovarian Cancer—Hyperthermia versus Normothermia: A Randomized Controlled Trial"

_jcm, 2022, doi:10.3390/jcm11195785_

Round 1

Reviewer 1 Report

The authors have described the results of their randomized controlled trial of IP chemo with paclitaxel and compared hypothermic and normothermic conditions.

HIPEC has been controversial since ages but have been tried by various clinicians with mixed results. 

The authors here have provided data that shows no significant difference in effects of hyperthermia and normothermia conditions. Reduction in cell cycle and cell proliferation markers by intra-peritoneal paclitaxel is encouraging.

This data adds to the information that will assist in determining whether HIPEC should be considered and improved in future.

I recommend accepting this manuscript with minor corrections to fix typos and text errors.

Author Response

To Editor and reviewers of Annals Surgical Oncology,

We are so grateful for the favorable comments about our manuscript entitled:

Role of Intraperitoneal Intraoperative Chemotherapy with Paclitaxel in the Surgical Treatment of Peritoneal Carcinomatosis from Ovarian Cancer. Hyperthermia versus Normothermia. A randomized controlled trial

 We have taken into account the many interesting observations made by you , thereby improving the quality of our study.

Reviewer 2 Report

The authors report on a small phase II randomized trial on the impact of hyperthermia during intraoperative intraperitoneal chemotherapy with paclitaxel in ovarian cancer patients. The presence or not of hyperthermic enhancement of paclitaxel is an interesting topic. The results of the study are limited due to the relatively small number of patients. Although there were no significant differences in pharmacokinetics, pathological response and expression of various markers, the authors mistakenly attempt to conclude that results with hyperthermic are better than normothermic intraoperative intraperitoneal chemotherapy.

In the Results of the Abstract, “Intraperitoneal PTX reduced the expression of p53, 98 p27, p21, ki67 and PCNA more in hyperthermia group, but not significantly.” should be replaced by mentioning that there were no statistically significant differences between the normothermic and the hyperthermic group, as stated before for pathological response and apoptosis.

The Introduction is not very well written. The authors state that cytoreductive surgery (CRS) and systemic chemotherapy is standard treatment for recurrent ovarian cancer. This is incorrect. The data from the three recently published randomized trials were inconclusive regarding the benefit of secondary CRS, with the GOG-0213 reporting no difference in progression-free survival (PFS) and overall survival (OS), the SOC-study improved PFS and at interim analysis no difference in OS and the DESKTOP III trial improved PFS and OS. Patient selection seemed to be the main factor for improved outcome. Hence, I would suggest to state that CRS for recurrent disease may be beneficial in selected patients.

The higher intraperitoneal (IP) concentrations are not due to the direct administration, but due to the slow absorption by the peritoneum and the first-pass effect by the liver which allows for higher intraperitoneal doses and potentially increased efficacy with concurrently low systemic toxicity. It does not result automatically in improved survival in all patients undergoing IP chemotherapy, so I would suggest to leave it to ‘potentially’ improves survival. Please add that because the penetration depth of IP delivered drugs into the tumor nodules is limited, optimal CRS is required before IP chemotherapy. Please clarify that IP chemotherapy is not widely accepted due its peritoneal access device complications, high toxicity in the last randomized trial (reference 5) and the complex and demanding logistical management of the patients. Moreover, another point of criticism is that in the randomized IP chemotherapy trials ovarian cancer patients in the control arm received lower doses and outdated schemes and in the last GOG-252 trial no improvement with IP chemotherapy was seen.

The main advantage of intraoperative use if IP chemotherapy is that microscopic (or small volume) disease is treated before regrowth occurs, that there is a better distribution of the drug solution to the entire seroperitoneal surface due to the lack of postoperative adhesions and the ability to perform it under hyperthermic conditions that are not tolerated by awake patients.

The authors should explain why it is important to know whether normothermic is as cytotoxic or less cytotoxic than hyperthermic intraoperative intraperitoneal chemotherapy. In case there is no difference, the adverse effects of hyperthermia (see: de Bree E, Katsougkri D, Polioudaki H, et al. Hyperthermia during intraperitoneal chemotherapy with paclitaxel or docetaxel for ovarian cancer: is there any benefit? Anticancer Res. 2020 Dec;40(12):6769-6780) can be avoided.

In the Patients and Methods section, should be feasibility to perform optimal or complete CRS an inclusion criterium? Why were serum concentrations also measured and not only plasma concentrations as indication for systemic exposure?

In the Results is stated that hematological toxicity was 18.8%, 12.5% (2/16) in the H-group and 6.3% (1/16) in the N-group. The hematological toxicity for the entire group should not be the sum of that for the H- and N-groups, but 9.4% (3/32).

Please state in the text which grade of pathological response was observed for both groups and refer to table 3.

At the end of the Results section, since there are no significant differences and values have a huge variety “Although no statistically significant, our results suggest that there is a tendency to decrease the cell cycle and proliferation markers in H-group” is incorrect to be stated and should be deleted. The conclusion should be that no differences could be demonstrated. Figure 2 should not be referred to as demonstrating the difference between both groups. It just shows one patient and should be inserted earlier in the paragraph.

In the Discussion is stated that CRS and HIPEC is standard treatment for colorectal cancer with peritoneal metastases. This is not the case after the French PRODIGE 7 randomized trial, which showed no benefit for CRS and HIPEC over CRS only.

Regarding the evidence for the benefit of HIPEC for ovarian cancer, a more recent meta-analysis may be referred to (Kim SI, Cho J, Lee EJ, et al. Selection of patients with ovarian cancer who may show survival benefit from hyperthermic intraperitoneal chemotherapy: A systematic review and meta-analysis. Medicine (Baltimore). 2019 Dec;98(50):e18355). Moreover, two recent randomized trials on HIPEC during interval CRS for primary ovarian cancer should be referred to (van Driel WJ, Koole SN, Sikorska K, et al. Hyperthermic intraperitoneal chemotherapy in ovarian cancer. N Engl J Med. 2018 Jan 18;378(3):230-240 and Lim MC, Chang SJ, Park B, et al. Survival after hyperthermic intraperitoneal chemotherapy and primary or interval cytoreductive surgery in ovarian cancer: a randomized clinical trial. JAMA Surg. 2022 May 1;157(5):374-383.). The study of Spiliotis et al. on CRS and HIPEC for recurrent ovarian cancer which is referred to has raised some questions as for its validity: the randomization process was not described in detail, primary end points were not clearly defined, there was no information provided regarding DFS, complications, postoperative systemic chemotherapy and follow-up, and the study had not been registered in an international clinical trial database (Harter P, Reuss A, Sehouli J, Chiva L, du Bois A. Brief report about the role of hyperthermic intraperitoneal chemotherapy in a prospective randomized phase 3 study in recurrent ovarian cancer from Spiliotis et al. Int J Gynecol Cancer. 2017 Feb;27(2):246-247). Moreover, others raised that the statistical analysis performed in the study was not clearly described and inappropriately applied, mean instead of median OS was used, reported data were inconsistent with provided graphics and their recalculation of the statistics demonstrated the outcome after HIPEC to be not statistically significantly superior to the control group (Batista TP. Comment on: surgery and HIPEC in recurrent epithelial ovarian cancer: a prospective randomized phase III study. Ann Surg Oncol. 2017 Dec;24(Suppl 3):630; Sanz Rubiales Á, Del Valle ML. Survival analysis in a randomized trial of HIPEC in ovarian cancer. Ann Surg Oncol. 2017 Dec;24(Suppl 3):631). This should be shortly noted.

In the Discussion should also be discussed a recent in vitro study on the effect of hyperthermia on the cytotoxicity of paclitaxel (de Bree E, Katsougkri D, Polioudaki H, et al. Hyperthermia during intraperitoneal chemotherapy with paclitaxel or docetaxel for ovarian cancer: is there any benefit? Anticancer Res. 2020 Dec;40(12):6769-6780).

At the end of the Discussion, is stated ‘Although no significant, our results suggest that there is a greater decrease in these tumor markers in H-group: a fact that could be related to the cytotoxic effect "per se" of hyperthermia.’. Once again, without significant differences and with such a wide variety (confidence intervals) in values it is incorrect to just suggest there is a difference and that this is due to the direct effect of hyperthermia. Please delete this sentence.

Similarly for the last sentence of the conclusion ‘Although no significant 324 differences were found between the two groups, it seems that hyperthermia may have a negative effect on the pharmacokinetics of PTX, but could enhance its cytotoxic effect, making this combination an effective treatment.’ This is not supported by the results of the study. The conclusion should be that in this randomized study, limited by the small number of patients included, no significant difference in pharmacokinetics and cytotoxicity could be demonstrated between normothermic and hyperthermic intraoperative intraperitoneal chemotherapy in patients with primary or recurrent ovarian cancer.

Linguistic review is definitely required.

Author Response

Response to the reviewer,

Thanks in advance for your revisión, it will improve the quality of our study.

In the Results of the Abstract, “Intraperitoneal PTX reduced the expression of p53, 98 p27, p21, ki67 and PCNA more in hyperthermia group, but not significantly.” should be replaced by mentioning that there were no statistically significant differences between the normothermic and the hyperthermic group, as stated before for pathological response and apoptosis.

I agree with the comment, it has been modified.

The Introduction is not very well written. The authors state that cytoreductive surgery (CRS) and systemic chemotherapy is standard treatment for recurrent ovarian cancer. This is incorrect. The data from the three recently published randomized trials were inconclusive regarding the benefit of secondary CRS, with the GOG-0213 reporting no difference in progression-free survival (PFS) and overall survival (OS), the SOC-study improved PFS and at interim analysis no difference in OS and the DESKTOP III trial improved PFS and OS. Patient selection seemed to be the main factor for improved outcome. Hence, I would suggest to state that CRS for recurrent disease may be beneficial in selected patients.

I agree with the comment, this part has been deleted.

The higher intraperitoneal (IP) concentrations are not due to the direct administration, but due to the slow absorption by the peritoneum and the first-pass effect by the liver which allows for higher intraperitoneal doses and potentially increased efficacy with concurrently low systemic toxicity. It does not result automatically in improved survival in all patients undergoing IP chemotherapy, so I would suggest to leave it to ‘potentially’ improves survival. Please add that because the penetration depth of IP delivered drugs into the tumor nodules is limited, optimal CRS is required before IP chemotherapy. Please clarify that IP chemotherapy is not widely accepted due its peritoneal access device complications, high toxicity in the last randomized trial (reference 5) and the complex and demanding logistical management of the patients. Moreover, another point of criticism is that in the randomized IP chemotherapy trials ovarian cancer patients in the control arm received lower doses and outdated schemes and in the last GOG-252 trial no improvement with IP chemotherapy was seen.

It has been modified..

The authors should explain why it is important to know whether normothermic is as cytotoxic or less cytotoxic than hyperthermic intraoperative intraperitoneal chemotherapy. In case there is no difference, the adverse effects of hyperthermia (see: de Bree E, Katsougkri D, Polioudaki H, et al. Hyperthermia during intraperitoneal chemotherapy with paclitaxel or docetaxel for ovarian cancer: is there any benefit? Anticancer Res. 2020 Dec;40(12):6769-6780) can be avoided.

I agree with this comment and this has been referred in the discussion section.

In the Patients and Methods section, should be feasibility to perform optimal or complete CRS an inclusion criterium? Why were serum concentrations also measured and not only plasma concentrations as indication for systemic exposure?

The serum concentration of Paclitaxel has been stablished by our group better than plasma. Fernández-Peralbo MA, Priego-Capote F, Luque de Castro MD, Casado-Adam A, Arjona-Sánchez A, Muñoz-Casares FC. LC–MS/MS quantitative analysis of paclitaxel and its major metabolites in serum, plasma and tissue from women with ovarian cancer after intraperitoneal chemoterapy. J Pharm Biomed Anal 2014; 91:131–7.

In the Results is stated that hematological toxicity was 18.8%, 12.5% (2/16) in the H-group and 6.3% (1/16) in the N-group. The hematological toxicity for the entire group should not be the sum of that for the H- and N-groups, but 9.4% (3/32).

Thanks, that has been modified.

Please state in the text which grade of pathological response was observed for both groups and refer to table 3.

I has been included

At the end of the Results section, since there are no significant differences and values have a huge variety “Although no statistically significant, our results suggest that there is a tendency to decrease the cell cycle and proliferation markers in H-group” is incorrect to be stated and should be deleted. The conclusion should be that no differences could be demonstrated. Figure 2 should not be referred to as demonstrating the difference between both groups. It just shows one patient and should be inserted earlier in the paragraph.

Agree and modified.

In the Discussion is stated that CRS and HIPEC is standard treatment for colorectal cancer with peritoneal metastases. This is not the case after the French PRODIGE 7 randomized trial, which showed no benefit for CRS and HIPEC over CRS only.

Modified

Regarding the evidence for the benefit of HIPEC for ovarian cancer, a more recent meta-analysis may be referred to (Kim SI, Cho J, Lee EJ, et al. Selection of patients with ovarian cancer who may show survival benefit from hyperthermic intraperitoneal chemotherapy: A systematic review and meta-analysis. Medicine (Baltimore). 2019 Dec;98(50):e18355). Moreover, two recent randomized trials on HIPEC during interval CRS for primary ovarian cancer should be referred to (van Driel WJ, Koole SN, Sikorska K, et al. Hyperthermic intraperitoneal chemotherapy in ovarian cancer. N Engl J Med. 2018 Jan 18;378(3):230-240 and Lim MC, Chang SJ, Park B, et al. Survival after hyperthermic intraperitoneal chemotherapy and primary or interval cytoreductive surgery in ovarian cancer: a randomized clinical trial. JAMA Surg. 2022 May 1;157(5):374-383.). The study of Spiliotis et al. on CRS and HIPEC for recurrent ovarian cancer which is referred to has raised some questions as for its validity: the randomization process was not described in detail, primary end points were not clearly …….

The discussion has been up to date don this issue.

In the Discussion should also be discussed a recent in vitro study on the effect of hyperthermia on the cytotoxicity of paclitaxel (de Bree E, Katsougkri D, Polioudaki H, et al. Hyperthermia during intraperitoneal chemotherapy with paclitaxel or docetaxel for ovarian cancer: is there any benefit? Anticancer Res. 2020 Dec;40(12):6769-6780).

It has been added

At the end of the Discussion, is stated ‘Although no significant, our results suggest that there is a greater decrease in these tumor markers in H-group: a fact that could be related to the cytotoxic effect "per se" of hyperthermia.’. Once again, without significant differences and with such a wide variety (confidence intervals) in values it is incorrect to just suggest there is a difference and that this is due to the direct effect of hyperthermia. Please delete this sentence.

Deleted

Similarly for the last sentence of the conclusion ‘Although no significant 324 differences were found between the two groups, it seems that hyperthermia may have a negative effect on the pharmacokinetics of PTX, but could enhance its cytotoxic effect, making this combination an effective treatment.’ This is not supported by the results of the study. The conclusion should be that in this randomized study, limited by the small number of patients included, no significant difference in pharmacokinetics and cytotoxicity could be demonstrated between normothermic and hyperthermic intraoperative intraperitoneal chemotherapy in patients with primary or recurrent ovarian cancer.

Totally agree and modified

Linguistic review is definitely required

Edited

Reviewer 3 Report

This work covers an interesting topic in locoregional treatment, aiming to analyse the role of hyperthermia in HIPEC with taxane in ovarian advanced cancer. Rationale, aim, statistical methods and lab analysis are logical and methodology is robust. It has been interesting reading the paper and it is concise with clear results.

I have no major concerns about this paper, just a few minor requests and some suggestions to possibly improve work quality.

There are some errors in spelling (probably coming from English-Spanish similar words), such as CASPASA-3 that should be caspase-3 (page 4 line 193), neoadjuvancia (page 6 line 298), and longer rather than higher same page line 290. I suggest a minor linguistic revision.

In addition to this, I would like to have some explanations from Authors and to give suggestions:

-sample size: Authors hypothesize that hyperthermia increase tumor regression by 39% (1% response in control and 40% in exp arm). First: tumor regression should be quantified by a scale/classification, so it would be useful to know grade of regression (e.g.: at least 1 level on a 4/5 tiers classification or complete response…) and if regression is based on best/worst specimen or an average of specimen. Second: how have Authors chosen an increase of 39%? Do they perform a pilot study or there are data in literature? Threshold values should be justified, since in my opinion this is the main cause of the missed statistical relevance of immunohistochemical analysis. Seeing tendency and values of p53, caspase (and so on), seems that hyperthermia increases tumor apoptosis (as it would be logical), but p values are >0.05, probably because sample is too small due to a too big expected effect of hyperthermia (40% of increase just by heat probably is too much). It is clear that threshold value is a balance between statistical power, known data and feasibility of the study, so Authors could have decided this threshold to permit enrolment of a reasonable number of patients. This point should also be addressed in discussion

-vast majority of patients received interval CRS+HIPEC, since systemic taxane can induce resistance also to IP chemotherapeutics, it would be possible to do a sub-analysis according to pre-op number of cycles? If this hypothesis would be correct, patients receiving 4 cycles should have a better immunohistochemical response compared to patients receiving 6 cycles (that should harbour more resistant cells)

-do the Authors have performed some kind of IP taxane spatial distribution analysis? Hyperthermia is thought to improve tissue perfusion of IP drugs, so it could be possible that experimental arm has a deeper drug perfusion, that can be considered an advantage of technique (according to Ansaloni 0.5 mm is the mean depth of taxane perfusion in tissue). So, Author could perform an analysis comparing concentration of taxane 0.5 mm from surface in the two arms and see if difference is significant. Frankly, I do not know if it would be possible to do it on paraffin embedded samples. This point is more a suggestion for Authors, than a request of explanation

-another interesting point is the heat-related damage on tissues and normal proteins. Do Authors have some data on that? For example: pre-post CRS+HIPEC and 1-2 post-operative days values of albumin in the two groups? Analysis of albumin that is routinely performed in these patients, could give some idea of heat-related tissue damage.

Author Response

To Editor and reviewers of Annals Surgical Oncology,

We are so grateful for the favorable comments about our manuscript entitled:

Role of Intraperitoneal Intraoperative Chemotherapy with Paclitaxel in the Surgical Treatment of Peritoneal Carcinomatosis from Ovarian Cancer. Hyperthermia versus Normothermia. A randomized controlled trial.

 We have taken into account the many interesting observations made by the reviewers, thereby improving the quality of our study.Thank you for considering our study suitable for publication, qualifying it as “Rationale, aim, statistical methods and lab analysis are logical and methodology is robust. It has been interesting reading the paper and it is concise with clear results ".

The spelling errors have been solved. 

Sample size calculation according to the public grant founding has been added in methods section. 

"Vast majority of patients received interval CRS+HIPEC, since systemic taxane can induce resistance also to IP chemotherapeutics, it would be possible to do a sub-analysis according to pre-op number of cycles? If this hypothesis would be correct, patients receiving 4 cycles should have a better immunohistochemical response compared to patients receiving 6 cycles (that should harbour more resistant cells)" I think that this is an interesting issue to perform in future exploratory analysis with a higher cohort that we have outside the clinical trial. Unfortunately , the trial is over and we have not financial support for this. 

do the Authors have performed some kind of IP taxane spatial distribution analysis? Hyperthermia is thought to improve tissue perfusion of IP drugs, so it could be possible that experimental arm has a deeper drug perfusion, that can be considered an advantage of technique (according to Ansaloni 0.5 mm is the mean depth of taxane perfusion in tissue). So, Author could perform an analysis comparing concentration of taxane 0.5 mm from surface in the two arms and see if difference is significant. Frankly, I do not know if it would be possible to do it on paraffin embedded samples. This point is more a suggestion for Authors, than a request of explanation

-another interesting point is the heat-related damage on tissues and normal proteins. Do Authors have some data on that? For example: pre-post CRS+HIPEC and 1-2 post-operative days values of albumin in the two groups? Analysis of albumin that is routinely performed in these patients, could give some idea of heat-related tissue damage.

I am completely agree that these analysis could be of interest on this trial. As I explained above the trial is over and founding ended. But I think that this interesting suggestion will be take in mind in future analysis.

Reviewer 4 Report

A very interesting study with a limited number of patients, but studies comparing normothermic versus hyperthermic HIPEC in patients are rare. The authors did not find differences in the immune/tumor markers they determined, and even that result is worth reporting provided the authors address and mention a few important issues.

In that light I would have expected that the authors would also discuss other studies with similar design. The authors should for instance also refer to another elegant 3-arm randomized clinical trial from Japan where the authors compared normothermic versus hyperthermic HIPEC in 139 patients with T2-4 gastric cancer, where HIPEC was performed with MMC. That study demonstrated that normothermic HIPEC did not give a benefit compared to no HIPEC, whereas HIPEC at 42°C did give a benefit. See:

Y. Yonemura, X. de Aretxabala, T. Fujimura, S. Fushida, K. Katayama, E. Bandou, K. Sugiyama, T. Kawamura, K. Kinoshita, Y. Endou, T. Sasaki. Intraoperative chemohyperthermic peritoneal perfusion as an adjuvant to gastric cancer: final results of a randomized controlled study. Hepatogastroenterology, 48 (2001), pp. 1776-1782

I would also expect the authors would provide more detail why they included so few patients. The low number of patients can be responsible for the fact that some trends in outcome did not get near significance, like the reported microscopically complete cytoreduction (CC0)  achieved in 81.2% in the Hyperthermic-group versus 62.5% in the Normothermic group. A nearly 20% difference, so that failure to establish significance can be attributed  to the use of just 16 patients per arm, so 32 patients in total. NB text has typo: ‘16 patients pre group’ instead of ‘16 patients per group’ (page 4, line 156). I know that the authors were mainly aiming at identifying markers, but they will agree with me that differences in markers are likely associated with differences in outcome, so require higher patient numbers if you want to be able to link changes in markers to differences in outcome.

The authors refer to the great variety in schedules and agents used for peritoneal metastases of ovarian origin (ref 16). That is true, the group of the authors believe in the use of taxanes because these are also effective in systemic use. They should also mention in the discussion that some other groups base their choice of agent and schedule more on known thermal enhancement of the agent,  like for instance cisplatin. Some papers present a good overview of the different thermal enhancement ratios for different drugs, e.g. by Issels:

Issels RD. Hyperthermia adds to chemotherapy. Eur J Cancer. 2008 Nov;44(17):2546-54

When this thermal enhancement on molecular scale has been established, and also their effective clinical use, see for instance the positive randomized trial of van Driel et al:

van Driel WJ, Koole SN, Sikorska K, Schagen van Leeuwen JH, Schreuder HWR, Hermans RHM, de Hingh IHJT, van der Velden J, Arts HJ, Massuger LFAG, Aalbers AGJ, Verwaal VJ, Kieffer JM, Van de Vijver KK, van Tinteren H, Aaronson NK, Sonke GS. Hyperthermic Intraperitoneal Chemotherapy in Ovarian Cancer. N Engl J Med. 2018 Jan 18;378(3):230-240

The authors do refer to the positive randomized trial of Spiliotis (page 6, line 261-264), but without mentioning the schedule and the agent used by Spiliotis et al. Please add that information too as this is relevant.

Question is whether there is evidence for direct enhancement of the effect of paclitaxel by heat, and whether hyperthermia combined with paclitaxel can impact the immune system. A recent study from Taiwan did show that hyperthermia can indeed increase the immune mitigated effect of chemotherapy, paclitaxel versus cisplatin. Those authors performed an elegant study in immune competent versus immune deficient mice, the authors should refer to this interesting study:

Chao-Chih Wu, Yun-Ting Hsu & Chih-Long Chang, Hyperthermicintraperitoneal chemotherapy enhances antitumor effects on ovarian cancer through immunemediated cancer stem cell targeting, International Journal of Hyperthermia, 2021, 38:1, 1013-1022

I also think the authors should refer to a recent review by van Rhoon et al on clinical evidence for hyperthermia in which the authors also discuss the degree of thermal enhancement reported for different agents used in cancer treatment:

van Rhoon GC, Franckena M, Ten Hagen TLM. A moderate thermal dose is sufficient for effective free and TSL based thermochemotherapy. Adv Drug Deliv Rev. 2020;163-164:145-156

Author Response

Thanks for your valuable comments about our trial. These will improve the quality of our study.

We have reviewed the language.

I have added the reference to the RCT from Yonemura et al., that certainly showed and improvement with the use of hyperthermia.

The simple size calculation has been performed according to the founding from the public grant obtained, it has been included in methods.

Spiliotis data have been added to the text

The Van Driel study has been included as you suggest. As well as, I have included two more recent RCTs.

The other references suggested by you have been included in the discussion.

Round 2

Reviewer 2 Report

The authors stated after every comment that they have made all changes to the manuscript according to these comments, but in fact this is misleading since they did not do so for many comments.

In the Conclusions of the Abstract, please add ‘statistically significant’, so noting “without finding statistically significant differences”. As noted previously, differences there are, but they are not statistically significant.

In the Introduction, the authors did not correct the comment on the pharmacokinetics. “The higher intraperitoneal (IP) concentrations are not due to the direct administration, but due to the slow absorption by the peritoneum and the first-pass effect by the liver which allows for higher intraperitoneal doses and potentially increased efficacy with concurrently low systemic toxicity.” Neither had they added the fact that because the penetration depth of IP delivered drugs into the tumor nodules is limited, optimal CRS is required before IP chemotherapy. Neither they clarifed that IP chemotherapy is not widely accepted due its peritoneal access device complications, high toxicity in the last randomized trial (reference 5) and the complex and demanding logistical management of the patients. Moreover, another point of criticism is that in the randomized IP chemotherapy trials ovarian cancer patients in the control arm received lower doses and outdated schemes and in the last GOG-252 trial no improvement with IP chemotherapy was seen.

Once again, the authors should explain why it is important to know whether normothermic is as cytotoxic or less cytotoxic than hyperthermic intraoperative intraperitoneal chemotherapy. In case there is no difference, the adverse effects of hyperthermia (see: de Bree E, Katsougkri D, Polioudaki H, et al. Hyperthermia during intraperitoneal chemotherapy with paclitaxel or docetaxel for ovarian cancer: is there any benefit? Anticancer Res. 2020 Dec;40(12):6769-6780) can be avoided. The adverse effects of hyperthermia should be referred to.

Once again, in the Patients and Methods section, should be feasibility to perform optimal or complete CRS an inclusion criterium?

Once again, in the Results, Figure 2 should not be referred to as demonstrating the difference between both groups. It just shows one patient and should be inserted earlier in the paragraph.

In the Discussion, ‘The present study has not showed differences …’ should be ‘The present study has not showed statistically significant differences’.

Once again, the authors have not corrected their discussion accordingly to the comments on the proof for CRS and HIPEC. Instead of analyzing more the recent randomized trials on primary ovarian cancer, they analyzed the randomized trial on recurrent ovarian cancer, which has been criticized widely. The randomized trials on primary ovarian cancer are added briefly at the end, but should be discussed in the earlier paragraph in which the study of Spiliotis is discussed. Moerover, a more recent meta-analysis may be referred to (Kim SI, Cho J, Lee EJ, et al. Selection of patients with ovarian cancer who may show survival benefit from hyperthermic intraperitoneal chemotherapy: A systematic review and meta-analysis. Medicine (Baltimore). 2019 Dec;98(50):e18355). Once again, the study of Spiliotis et al. on CRS and HIPEC for recurrent ovarian cancer which is referred to has raised some questions as for its validity: the randomization process was not described in detail, primary end points were not clearly defined, there was no information provided regarding DFS, complications, postoperative systemic chemotherapy and follow-up, and the study had not been registered in an international clinical trial database (Harter P, Reuss A, Sehouli J, Chiva L, du Bois A. Brief report about the role of hyperthermic intraperitoneal chemotherapy in a prospective randomized phase 3 study in recurrent ovarian cancer from Spiliotis et al. Int J Gynecol Cancer. 2017 Feb;27(2):246-247). Moreover, others raised that the statistical analysis performed in the study was not clearly described and inappropriately applied, mean instead of median OS was used, reported data were inconsistent with provided graphics and their recalculation of the statistics demonstrated the outcome after HIPEC to be not statistically significantly superior to the control group (Batista TP. Comment on: surgery and HIPEC in recurrent epithelial ovarian cancer: a prospective randomized phase III study. Ann Surg Oncol. 2017 Dec;24(Suppl 3):630; Sanz Rubiales Á, Del Valle ML. Survival analysis in a randomized trial of HIPEC in ovarian cancer. Ann Surg Oncol. 2017 Dec;24(Suppl 3):631). This should be shortly noted.”

The study of Yonemura et al. (reference 69) should be deleted, since it does not refer to normothermic versus hyperthermic intraperitoneal chemotherapy in gastric cancer, but to HIPEC versus no HIPEC. It does not fit in the discussion.

Still linguistic review is definitely required.

Author Response

document upload

Reviewer 4 Report

the authors have addressed my comments in an acceptable fashion

Author Response

Thanks so much for your comment